

# A Versatile Vacuum Ultraviolet Ion Source for Reduced Pressure Bipolar Chemical Ionization Mass Spectrometry

Martin Breitenlechner[1,2], Gordon A. Novak[1,2], J. Andrew Neuman[1,2], Andrew W. Rollins[1] and Patrick R. Veres[1]

[1]NOAA Chemical Science Laboratory (CSL), 325 Broadway, Boulder, Colorado 80305, USA
[2]Cooperative Institute for Research in Environmental Sciences (CIRES), University of Colorado Boulder, Boulder, Colorado 80309, USA

*Correspondence to*: Patrick (Patrick.Veres@noaa.gov)

**Abstract**

We present the development of a Chemical Ionization Mass Spectrometer (CIMS) ion source specifically designed for in situ measurements of trace gases in the upper troposphere and lower stratosphere. The ion source utilizes a commercially available photoionization krypton lamp, primarily emitting photons in the vacuum ultraviolet (VUV) region at wavelengths of 124 and 117 nm (corresponding to energies of 10 and 10.6 eV, respectively), coupled to a commercially available Vocus Proton Transfer Reaction Mass Spectrometer. The VUV ion source can produce both negative and positive reagent ions, however, here we primarily focus on generating iodide anions ($I^-$). The instrument's drift tube (also known as ion molecule reactor) operates at pressures between 2 and 10 mbar, which facilitates ambient sampling at atmospheric pressures as low as 50 mbar. The low operating pressure reduces secondary ion chemistry that can occur in iodide CIMS. It also allows the addition of water vapor to the drift tube to exceed typical ambient humidity by more than one order of magnitude, significantly reducing ambient humidity dependence of sensitivities. An additional benefit of this ion source and drift tube is a 10 to 100-fold reduction in nitrogen consumed during operation relative to standard $I^-$ ion sources, resulting in significantly reduced instrument weight and operational costs. In iodide mode, sensitivities of 76 cps/ppt for nitric acid, 35 cps/ppt for $Br_2$, and 8.9 cps/ppt for $Cl_2$ were achieved. Lastly, we demonstrate that this ion source can generate benzene ($C_6H_6^+$) and ammonium ($NH_4^+$) reagent ions to expand the number of detected atmospheric trace gases.

## 1 Introduction

Chemical ionization mass spectrometry (CIMS) has been widely used as a powerful tool to detect atmospheric compounds present at trace levels (de Gouw and Warneke, 2007; Huey, 2007). In CIMS, reagent ions undergo mostly non-dissociating ion-molecule reactions in the ion molecule reactor (IMR). The resulting product ions are detected by a mass spectrometer. Modern instruments typically utilize time-of-flight mass spectrometers (ToF-MS) due their high mass resolving power capable of separating many isobars and simultaneous detection of ions over a wide mass range. The CIMS technique allows for accurate detection of atmospheric trace gases with high sensitivity, low detection limits and fast time responses. Additionally, CIMS is a highly customizable measurement technique that allows for the detection of a wide range of analyte species through reagent ion selection, utilizing both positive and negative reagent ions.

Common negative ion chemical ionization schemes include nitrate (Eisele and Tanner, 1993), trifluoromethoxy anion ($CF_3O^-$; Crounse et al., 2006), acetate (Veres et al., 2008; Bertram et al., 2011) and iodide anions (Caldwell et al., 1989; Slusher et al., 2004; Lee et al., 2014). Relative to nitrate ion CIMS, which is highly selective to sulfuric acid (Jokinen et al., 2012), amines (Simon



et al., 2016) and highly oxygenated molecules (Ehn et al., 2014), $CF_3O^-$, acetate or iodide are less selective and therefore are typically operated at reduced pressures to avoid depletion of primary ions and secondary ion chemistry. Iodide anions in particular have been used for the detection of both organic and inorganic acids (Roberts et al., 2010), including reactive nitrogen species (HNO$_3$, HONO, HO$_2$NO$_2$, peroxy acetyl nitrates, N$_2$O$_5$) and halogens (ClNO$_2$, HCl, BrO, HOBr, HOCl, Cl$_2$, Br$_2$) (Neuman et al., 2010; Lee et al., 2014; Liao et al., 2014; Lee et al., 2018; Slusher et al., 2004).

Positive ion CIMS ionization schemes include Proton-Transfer-Reaction mass spectrometry (PTR-MS, Hansel et al., 1995), which detects a wide range of organic compounds utilizing proton transfer reactions from hydronium ions, or more selective ionization schemes such as $O_2^+$ (Norman et al., 2007), $NO^+$ (Karl et al., 2012), ammonium reagent ions (NH$_4^+$, Zaytsev et al., 2018, Müller et al., 2020) and benzene ions (Leibrock and Huey, 2000; Kim et al., 2016; Lavi et al., 2018).

The primary goal of this work is to leverage the flexibility of the CIMS technique to develop an instrument capable of detecting a wide range of molecules in the upper troposphere and lower stratosphere (UT/LS) onboard a research aircraft which can operate up to 20 km altitude. The work presented here focuses on detailing the design and performance of a new ion source and instrument configured to meet several key challenges related to deployment of CIMS in the UT/LS: 1) flexibility in target analyte species, 2) sampling from pressure as low as 50 mbar, and 3) secondary ion chemistry anticipated due to the sample matrix. The specific challenges related to engineering a research grade instrument for operation in a low-pressure environment will be addressed in a future manuscript. Here we provide details on the design and performance of the ToF-MS instrument and our custom designed VUV ion source. Our results primarily focus on iodide ion reagent ions that are chosen to provide UT/LS observations of reactive nitrogen species and halogens. We present signal intensities, cluster distributions, sensitivities, limits of detection and an assessment of potential secondary ion chemistry. Additionally, we illustrate the versatility of the VUV ion source by highlighting its ability to generate benzene ions ($C_6H_6^+$) and ammonium ions (NH$_4^+$).

## 2 Materials and methods

The CIMS instrumentation used in this work was chosen to address the coupled issues of reduced pressure operation and interferences from secondary ion chemistry. Aircraft sampling inlets are commonly pressure controlled to maintain a pressure well below the minimum ambient pressure to be sampled (Veres et al. 2020) so that instrument operating conditions are insensitive to changes in aircraft altitude. High altitude aircraft that measure in the UT/LS can reach ambient pressures as low as 50 mbar. This pressure regime presents a challenge to standard iodide CIMS instrumentation that typically operates with an IMR pressure between 30 and 100 mbar. These pressures are necessary in order to maximize sensitivities by optimizing the number of collisions between reagent ions and reactants. However, pressure dependent secondary ion reactions can make interpretation of the observed mass spectra difficult, potentially leading to the misidentification of product ion species. For example, ozone levels exceed 1000 ppb in the LS, which can promote secondary ion chemistry originating from reagent ions being transformed by ozone in the IMR (Dörich et al., 2021; Zhang and Zhang, 2021; Section 3.3). Unfortunately, limiting secondary reactions requires a reduction in operating pressure, which will have an adverse impact on instrument sensitivities and detection limits in a standard CIMS instrument.

The pressure regime and bipolar electronics of the Vocus PTR-ToF instrument (Model S, Tofwerk AG, Thun, Switzerland) are particularly well-suited for this work. In the Vocus, the IMR is a drift tube that operates at pressures typically between 2 and 4 mbar, ideal for both limiting secondary ion reactions as well as allowing for adequate pressure control at high altitude. To



compensate for the lower collision frequencies at these pressures, an ion focusing quadrupole ion guide surrounding the drift tube enhances the transmission efficiency from the drift tube into the mass spectrometer. While the commercially available Vocus instrument is designed to operate in positive ion mode utilizing $H_3O^+$ primary ions, bipolar electronics allow for implementation of negative ion modes such as iodide anion CIMS. This feature allows for a large range of target analyte species with minimal instrument modifications. Therefore, the Vocus was chosen as a platform to develop a CIMS that is highly sensitive and operates

at low pressures – which is both suitable for high altitude deployment and in environments of high ozone levels, where secondary ion chemistry is of concern.

## 2.1 Vocus PTR-ToF

The Vocus is primarily designed to operate as a PTR-CIMS using hydronium ions ($H_3O^+$) as reagent ions coupled to a drift tube as an IMR (Krechmer et al., 2018) to reduce clustering of hydronium ions with omnipresent water vapor molecules in the sample

gas. Proton transfer reactions take place in a resistive glass drift tube which is surrounded by a quadrupole ion guide. The glass drift tube is 100 mm long and provides an axial electric field which can be set between zero and 80 V/cm. The electric field effectively supresses cluster formation of hydronium ions with water molecules at typical operating conditions of 60 V/cm at 2 mbar (The reduced electric field, E/N, expressed in units of Townsend (Td) is calculated as follows: $\frac{E}{N} = 1.38 \cdot \frac{T\,[K]}{p\,[Pa]} \cdot E\,[\frac{V}{cm}]$; 60 V/cm at 2 mbar at 20 °C equals 121 Td). The surrounding quadrupole focuses ions in the center of the device, reducing ion losses

at the exit of the drift tube and effectively increasing ion transmission and sensitivity approx. 10- to 100-fold compared to classical PTR-MS drift tubes (Krechmer et al, 2018). The instrument is equipped with a positive DC glow discharge, producing hydronium primary cations ($H_3O^+$), using pure water vapor as a source gas (Chemical ionization gas, CI gas). Contrary to many earlier PTR-MS instruments (Hansel et a., 1995, de Gouw et al., 2003), water vapor is not removed from the system before mixing with the analyte gas in the drift tube (Krechmer et al., 2018). The resulting elevated water vapor concentration reduces the dependence of

sensitivity on ambient humidity (Krechmer et al., 2018) compared to previous PTR-MS instruments (Warneke et al., 2000). While the Vocus PTR is typically used in positive ion mode, it can be switched to negative polarity. However, the glow discharge ion source cannot efficiently produce negative ions, so a different ion source is required to operate in negative mode. The Vocus model S in its standard configuration has a mass resolving power of up to 5000 m/Δm and sensitivities of up to 10 cps/ppt in $H_3O^+$ mode (Tofwerk AG Vocus product website, 2021).

## 100 2.2 VUV lamps

In this work, the glow discharge ion source used in the Vocus PTR-TOF was replaced with two vacuum ultraviolet lamps (Restek photoionization lamp, model 108-BTEX, Restek Corporation, PA, USA). The lamps are powered by a 3W DC Voltage supply, initially providing 1500 VDC to ignite the lamp. The supply is current limited to 1 mA, resulting in a continuous operating voltage of approx. 350 V. The lamps primarily emit photons at wavelengths of 124 and 117 nm, corresponding to photon energies of 10

and 10.6 eV, respectively. Ji et al (2020) pioneered the use of VUV lamps to produce iodide ions using a mixture of nitrogen and methyl iodide as a CI gas. This approach is versatile since the emitted photons ionize a multitude of molecules, provided their ionization energy is below 10.6 eV. Examples include ammonia ($NH_3$), benzene ($C_6H_6$), toluene ($C_7H_8$) and xylene ($C_8H_{10}$) as well as some small alkenes such as propene ($C_3H_6$).

These photoionization lamps produce reagent ions similarly to glow discharge ion sources, but differences exist: In a glow

discharge, a "self-cleaning" mechanism converts all fragments that arise from electron impact ionization of components present in the CI gas into a suitable primary species. One well-known system where this process is effective is the generation of $H_3O^+$ ions in a glow discharge (with pure water vapor as a CI gas) in a PTR-MS instrument: All possible fragments of electron impact



ionization of $H_2O$ ($H^+$, $H_2^+$, $OH^+$, $O^+$) as well as $H_2O^+$ undergo subsequent reactions with water molecules to form $H_3O^+$ primary ions (Hansel et al., 1995). One advantage of the VUV ion source described here is the absence of fragment ion species in the ion

source: eg. in the case of benzene, the fragment having the lowest appearance energy (AE) is $C_6H_5^+$ with an AE of 12.90 eV. This energy is well above the most energetic photons emitted by the lamp (10.6 eV) and therefore no fragments are expected anywhere in the source. This "soft" primary ionization allows for mixing of source gases without the complication of having to take into account secondary reactions of fragments with any member of the CI gas. Another advantage of the VUV source and its "soft" ionization process is the absence of fast ions or electrons and subsequent sputtering processes in the ion source region, resulting in

increased longevity of the ion source region without the need of cleaning.

### 2.3 Ion source design and coupling to the Vocus drift tube

To sample trace gases in the atmosphere, the pressure must be reduced from ambient pressure to a few mbar in the drift tube, and this is accomplished using several stages of pumping. The sample gas inlet consists of a PFA tube (ID 7.5 mm). The inlet is connected to a pump through a custom-made butterfly valve with minimal pressure drop to allow for a sampling flow of up to 10

slpm (at sea level). A pressure control system comprising another custom-built butterfly valve keeps the pressure in the region between the two critical orifices at 40 mbar, which is sensed through a separate port. The low impedance pumping port and the butterfly valve of this pressure control system allow for a wide dynamic range of ambient pressures, ranging from ground-level to a simulated altitude of 20 km (50 mbar). As described in section 3.1, in iodide mode, the drift voltage is set to zero in order to maximize sensitivity. As a result, reaction times of ions with molecules present in the sample (and, consequently, sensitivities)

solely depend on the gas flow through the drift tube. Therefore, a second stage pressure control system, regulating flow through the drift tube, compensates for potential changes of pumping speeds with aircraft altitude and assures that the reaction time of ions (affecting sensitivities) does not change in the absence of an electric field.

Figure 1 shows the coupling of the inlet to the ion source comprised of the VUV lamp and the Vocus drift tube. In this figure, one VUV lamp is shown for simplicity, however, the current design utilizes a second VUV lamp mounted at 180º. The sampled air

enters the drift tube through a 12 mm long PEEK capillary (ID 0.5 mm, with a typical flow rate of approx. 100 sccm), which is surrounded by a stainless steel sleeve, preventing exposure of the sample gas to photons emitted by the VUV lamps. The CI gas is introduced with a typical flow rate of approx. 20 sccm shared between the lamps. The lower operating pressure of the Vocus allows for the reduction of the total ion source flow to 20 sccm, while previous $I^-$ CIMS operate at elevated IMR pressures and typically consume 0.5 to 2 slpm of a methyl iodide/$N_2$ mixture (Veres 2020, Lee et al. 2014). A separate inlet port for humidification supplies

pure water vapor from the headspace of a liquid water reservoir to the sample gas, typically between 10 and 20 sccm. This results in humidity levels in the drift tube that exceed typical ambient water vapor concentrations by at least an order of magnitude. As shown in Figure 2, this has the advantage of establishing a reagent ion cluster distribution (and subsequently sensitivities) that is largely independent of the sample humidity.



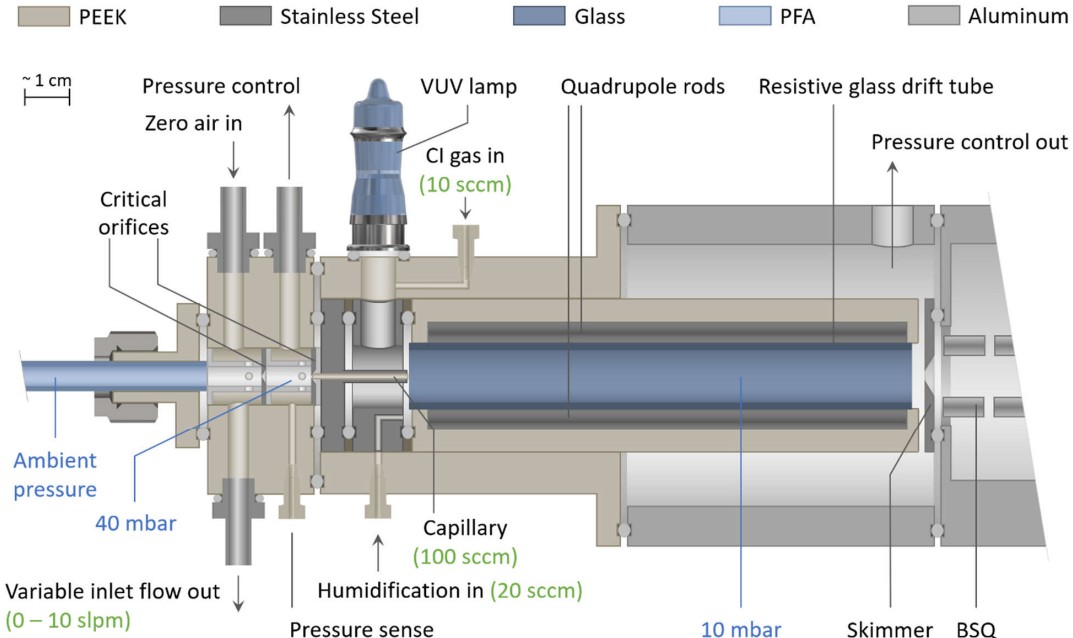


**Figure 1: The VUV ion source coupled to the Vocus PTR-ToF-MS. Note that only one VUV lamp is shown. Green numbers represent approximate typical flow rates.**

Details on the generation of reagent ions will be described in the following sections. Briefly, the CI gases used were (1) a custom-
made gas cylinder containing methyl iodide with a volume mixing ratio of 1000 ppm in nitrogen with a typical flow rate of typically
20 sccm for generating iodide ions; (2) a liquid reservoir of benzene at room temperature and ambient pressure which is overflown
by 10 to 20 sccm UHP nitrogen for generating benzene reagent ions; and (3) a similar setup with a 0.1% ammonium hydroxide
solution reservoir for generating ammonium ($NH_4^+$) reagent ions.

## 3 Results and discussion

### 3.1 Iodide ion CIMS

Iodide ions are commonly used for trace gas detection in CIMS instruments, and the measurement sensitivity and selectivity are
dictated by the conditions in the IMR/drift tube. Iodide adduct formation from I⁻ reagent ions can generally be described as follows:

$$I^- + R \rightarrow [I^- \bullet R]^* \xrightarrow{M} I^- \bullet R ,$$  (1)

where R is the sample molecule and the asterisk denotes a short-lived excited state, which can be stabilized via collisions with a
third body M. (Caldwell et al, 1989). The efficiency of collisional stabilization is pressure dependent and is generally less effective
at the lower operating pressures used in this work.

Iodide readily clusters with water, and the resulting cluster ions also serve as reagent ions for many species of interest. Therefore,
especially at low pressures, ligand switching reactions with iodide-water clusters leading to the same I⁻•R adduct ion are important:

$$I^- \bullet H_2O + R \rightarrow I^- \bullet R + H_2O .$$  (2)

These reactions are exothermic if the binding energy (BE) of I⁻•R exceed the BE of I⁻•$H_2O$ (43 kJ/mol, NIST Chemistry Webbook,
2021). The added humidification port (see Figure 1) provides the possibility to add pure water vapor, promoting the formation of



$I^-\bullet(H_2O)_n$ and adduct formations from reaction 2. Figure 2 shows the effect of adding water to the drift tube on the reagent ion cluster distribution. The distributions were calculated based on measured equilibrium constants (Enthalpy and Entropy changes) for the iodide-water system (Hiraoka et al., 1988, de Gouw et al., 2003). As shown in panel b), figure 2, ambient humidity has no

significant effect on the cluster distribution when water is directly added to the drift tube. In figure 2c), we show the calculated residence time of the $I^-\bullet H_2O$ cluster in the drift tube based on a drift velocity resulting from a reduced ion mobility of 1.6 cm$^2$/Vs (Kilpatrick, 1971). In the absence of an electric field (E/N = 0 Td), the reaction time is estimated based on the gas exchange rate; at elevated reduced electric fields, the reaction time decreases due to increasing ion drift velocity. The reaction time of ions with the sample gas is directly proportional to the sensitivity (when normalized to primary ion intensity – or, in absolute terms, when

primary ion intensity does not change with the applied electric field). In $I^-$ mode, the instrument is operated without an electric field applied over the length of the drift tube, to maximize sensitivity as discussed earlier. Figure 3 shows a mass spectrum obtained during a calibration experiment (formic acid and nitric acid added to zero-air). During this experiment, the drift tube pressure was 9 mbar, with 18 sccm of water vapor added.

Table 1 shows the observed reagent ion intensities for $I^-$ (m/z = 126.9050 Th), $I^-\bullet H_2O$ (m/z = 144.9156 Th) and $I^-\bullet(H_2O)_2$, (m/z =

162.9262 Th) and their respective distribution. Note that observed cluster distributions are expected to differ from the distributions in the IMR because the mass spectrum is affected by voltages in the ion transfer region between the drift tube and the mass analyser. These voltages were optimized for maximum ion intensities of product ions (which have stronger binding energies than the $I^-\bullet H_2O$ cluster) rather than being optimized for minimum collision activation of ions and avoiding de-clustering in this region.






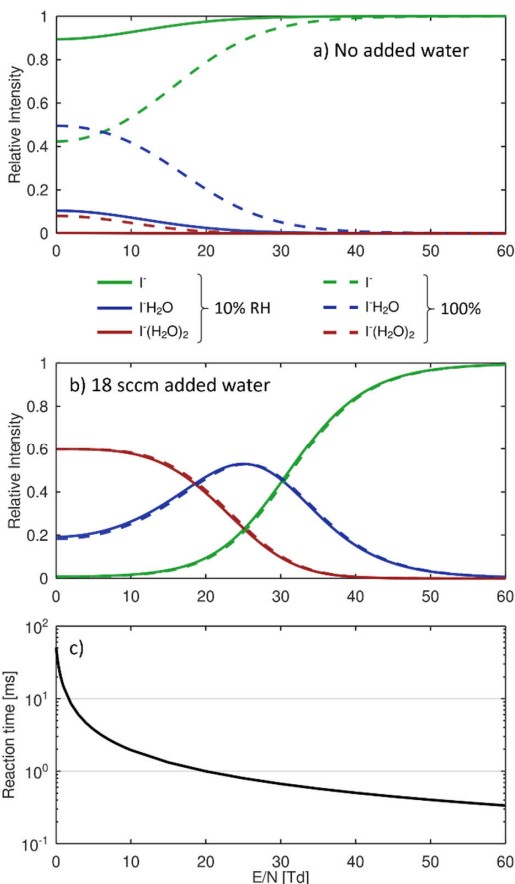

**Figure 2: Calculated cluster distributions based on enthalpy and entropy changes of association and dissociation reactions of the I⁻-water system. Solid lines show data for 100% relative humidity (RH) of the sample gas at 25 °C; dotted lines represent 10% RH at 25 °C. Panel a: no added water to the drift tube; Panel b: 18 sccm of water vapor added to the drift tube (approx. 1 mbar partial pressure); Panel c) Reaction time of I·H₂O as a function of reduced electric field (E/N).**




**Table 1: Reagent ion intensities in iodide mode**

|  |  | Ion intensities | |
| --- | --- | --- | --- |
| Ion formula | m/z [Th] | Cps [$10^6$ Hz] | percent |
| $I^-$ | 126.905 | 7.0 | 63 % |
| $I^- \cdot H_2O$ | 144.9156 | 3.8 | 34 % |
| $I^- \cdot (H_2O)_2$ | 162.9262 | 0.37 | 3 % |
| total |  | 11.2 | 100 % |




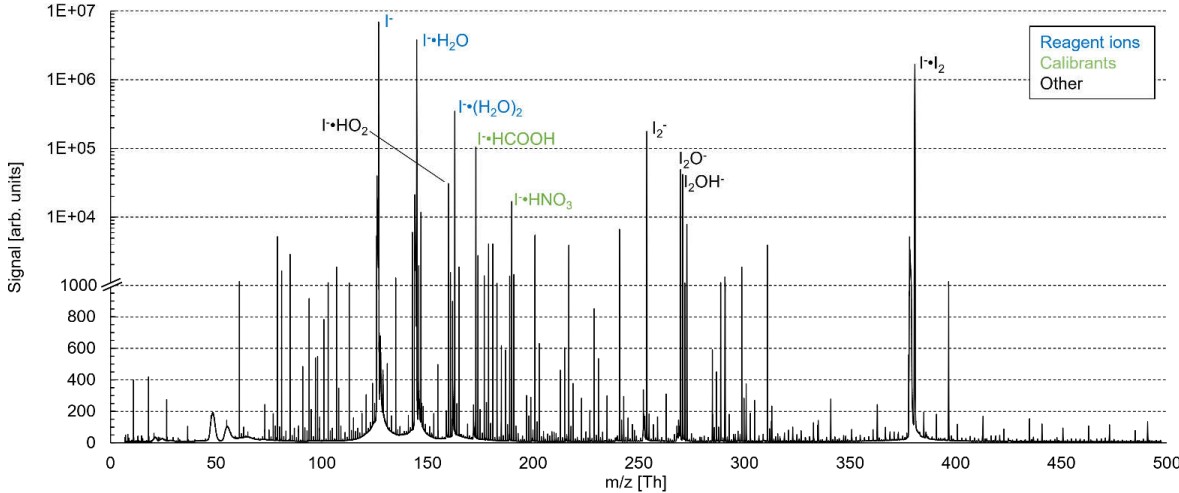


**Figure 3: Mass spectrum in iodide mode. Operating conditions: drift tube pressure: 9 mbar; H$_2$O addition: 20 sccm; calibrants: 14 ppb formic acid (HCOOH), 18 ppb nitric acid (HNO$_3$).**

## 3.2 Sensitivities and limits of detection

Instrument capabilities are assessed by admitting calibration standards of inorganic acids, organic acids, and halogens into the inlet. All analyses are performed using high resolution peak fits, and the resolution was typically 4500 m/Δm. Sensitivities of formic acid (I•HCOOH, m/z = 172.9105 Th), nitric acid (I•HNO$_3$, m/z = 189.9007 Th), bromine (I•Br$_2$, m/z = 284.7417 Th) and chlorine (I•Cl$_2$, m/z = 196.8427 Th) were obtained using permeation tubes, dynamically diluted with zero air. Nitryl chloride (I•ClNO$_2$, m/z = 207.8668 Th) was generated in a heterogenous reaction of Cl$_2$ and NO$_2$ as described in Thaler et al., 2011; The

reaction was monitored by measuring NO$_2$ (Ryerson et al., 2000). As shown in table 2, sensitivities range from 9 to 76 cps/ppt for these species. Limits of detection range from sub-ppt levels for 1 s measurements for bromine and chlorine to more than 50 ppt for formic acid, which is limited by a high background signal in the current setup. The sensitivities obtained with this instrument are similar to iodide CIMS instruments operating at much higher pressures and longer reaction times. Ji et al. (2020), operating the IMR between 26 and 53 mbar, report sensitivities in the range of 80 to 150 cps/ppt for formic acid, chlorine and nitryl chloride,

with limits of detections ranging from 0.1 ppt to 0.6 ppt for these species (1 minute integration time). Veres et al. (2020) operate their instrument at an IMR pressure of approx. 40 mbar, obtaining sensitivities on the order of 15 cps/ppt for the most sensitive compounds such as Cl$_2$, ClNO$_2$ and N$_2$O$_5$.

Figure 4 shows the humidity dependence of sensitivities for formic acid (HCOOH) and Bromine (Br$_2$), normalized to the driest conditions studied (RH = 30% at an ambient temperature of 22 °C), while 18 sccm of water vapor was directly added to the drift

tube. The change in sensitivities for both formic acid and bromine is less than 10% between 30% and 85% RH at room temperature. For comparison: Lee et al. (2014) reported that sensitivities increased by a factor of 30 for Br$_2$ and Cl$_2$, by a factor of 3 for HNO$_3$ and a decrease of 20% to 80% for OVOCs (C$_{10}$H$_{12-18}$O$_{2-9}$) compared to dry conditions when adding 0.8 mbar of water vapor to the IMR (corresponding to 38% RH at 20 °C).





**Table 2: Sensitivities and detection limits (for a sample interval of 10s, 3 standard deviations) for formic acid, nitric acid, bromine, chlorine and chlorine nitrite in iodide mode.**

| Species | Chemical formula | Sensitivity (cps/ppt) | Background (cps) | Background | Limit of detection 10 s, 3σ, (ppt) |
|---|---|---|---|---|---|
| Formic acid | $HCOOH$ | 23 | $7.9 \cdot 10^5$ | 34 ppb | 51 |
| Nitric acid | $HNO_3$ | 76 | $5.9 \cdot 10^4$ | 780 ppt | 4.3 |
| Bromine | $Br_2$ | 35 | 220 | 6 ppt | 0.6 |
| Chlorine | $Cl_2$ | 8.9 | 18 | 2 ppt | 0.7 |
| nitryl chloride | $ClNO_2$ | 9 | 72 | 8 ppt | 1.3 |


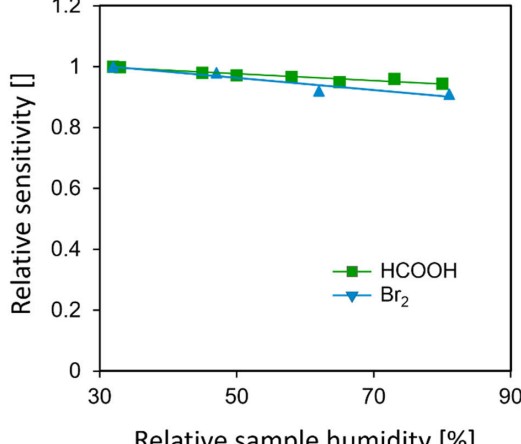

**Figure 4: Relative sensitivities of formic acid (HCOOH) and bromine ($Br_2$) as a function of sample relative humidity (RH; $T_{ambient}$ = 22°C, added $H_2O$ flow: 18 sccm). Sensitivities are normalized to the driest conditions (RH=30%).**


### 3.3 Secondary ion chemistry

The reaction of $I^-$ reagent ions with $O_3$ has the potential to drive secondary ion chemistry that produces interfering ions from $IO^-$
and $IO_3^-$ (Dörich et al., 2021; Zhang and Zhang, 2021). In the typical $I^-$ ionization mode, $I^-$ forms a stable adduct with the analyte (reaction 1). If ozone levels are sufficiently high, a fraction of $I^-$ reagent ions react with $O_3$ resulting in the production of both $IO^-$ and $IO_3^-$ (reactions 3 and 4).

$$I^- + O_3 + M \rightarrow IO_3^- + M \qquad (3)$$
$$IO_3^- + M \rightarrow IO^- + O_2 + M \qquad (4)$$

Secondary chemistry from $IO^-$ and $IO_3^-$ can form stable adducts with atmospheric species A, resulting in product ions that don't faithfully represent atmospheric composition:

$$IO^- + A \rightarrow IO^- \bullet A \qquad (5)$$
$$IO_3^- + A \rightarrow IO^- \bullet A + O_2 \qquad (6)$$



One example of an unwanted secondary ion chemistry effect described here affects the detection of HNO₃. Both IO₃⁻ and IO⁻ form

an artificial signal at IOHNO₃⁻ which would interfere with quantification of atmospheric peroxynitric acid (HNO₄):

$$IO^- + HNO_3 \rightarrow IO^- \bullet HNO_3 \tag{7}$$

$$IO_3^- + HNO_3 \rightarrow IO^- \bullet HNO_3 + O_2 \tag{8}$$

The secondary ion chemistry described here requires two sequential reactions, the first to form IO⁻ or IO₃⁻, which then react with the analyte in a second step. Therefore, operation of the drift tube at lower pressures should reduce secondary ion chemistry by

reducing the total number of collisions.

Laboratory calibrations show the decline of I⁻•HNO₃ and an increase of artificially detected I⁻•HNO₄ (from IO⁻•HNO₃) as a function of ozone mixing ratio at various reaction pressures (Fig. 5). A fixed HNO₃ mixing ratio was added during all experiments. Data for 40 and 93 mbar were measured with a standard I- CIMS (Veres et al, 2020). Our results show that a reduced operating pressure in Iodide CIMS reduces the potential of erroneous assignments of chemical species to peaks in the mass spectrum, although not

completely eliminating this possibility.

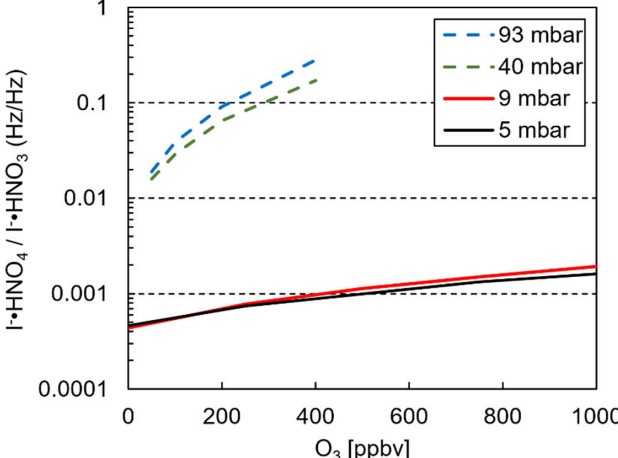

**Figure 5: Ratio of I⁻•HNO₄ and I⁻•HNO₃ as a function of ozone mixing ratio of the sample gas. Solid lines: Instrument described in this work operating at pressures indicated in the legend; Dashed lines: Data from a standard I⁻-CIMS operated at pressures indicated in the legend (40 and 93 mbar).**

## 4 Alternative reagent ion species

A variety of reagent ions of both polarities have been used to specifically detect different classes of compounds relevant in atmospheric chemistry. The VUV ion source is a versatile tool to produce different reagent ions of either polarity by using a suitable CI gas, with some limitations mainly arising from the photon energies from the source. Here we briefly describe experiments to demonstrate the capability of the VUV source to produce benzene ($C_6H_6^+$) and ammonium ($NH_4^+$) cations.





### 4.1 Benzene+ CIMS

Benzene cations have been shown to be both sensitive and selective reagent ions for chemical ionization of select biogenic volatile organic compounds, including dimethyl sulfide (DMS, Kim et al., 2016), isoprene (Lavi et al., 2018) and monoterpenes (Lavi et al., 2018). Benzene ($C_6H_6$) with an ionization potential of 9.24 eV (NIST Chemistry Webbook, 2021) is directly ionized by the photons originating from the lamps, producing $C_6H_6^+$ primary ions. The most energetic photons emitted by the lamp have energies of 10.6 eV, significantly lower than the lowest appearance energy of any fragment from the benzene molecule ($C_6H_5^+$ at 12.90 eV).

Therefore, no fragment ion species are expected, nor observed. Benzene is added to the instrument by flowing 10 to 20 sccm UHP nitrogen over a liquid reservoir (room temperature, ambient pressure), resulting in a benzene partial pressure of approx. 0.4 mbar in the source region, at an operating pressure of 3.8 mbar. Monoterpenes (α-pinene and limonene) and isoprene were detected at their respective adduct ions: $C_6H_6^+ \bullet C_{10}H_{16}$ (m/z = 214.1716 Th) for monoterpenes and $C_6H_6^+ \bullet C_5H_8$ (m/z = 146.1090 Th) for isoprene. No monoterpene fragment ion species were observed in the mass spectra. Sensitivities of 14 cps/pptv for α-pinene, 27

cps/pptv for limonene and 8.2 cps/pptv for isoprene were achieved, without significant humidity dependency, as shown in Figure 6b. The absence of fragmentation and humidity dependence observed here is a notable improvement over previous benzene CIMS instruments which report strong humidity dependence and a range of fragmentation products for various monoterpenes which complicates quantification with those instruments (Kim et al., 2016. Lavi et al., 2018).

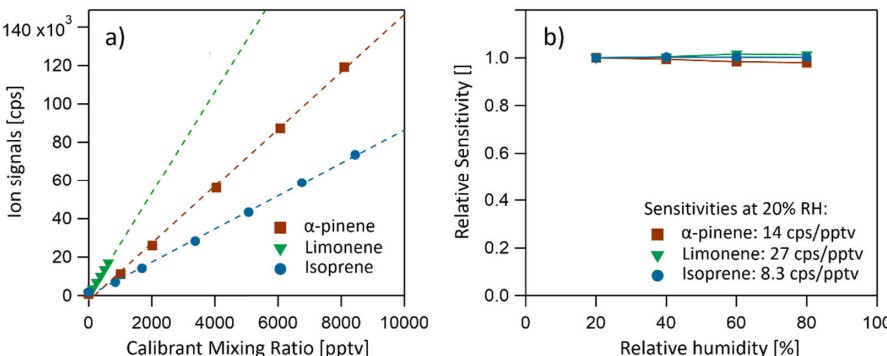


**Figure 6: Calibrations in benzene mode; panel a) instrument response to selected biogenic calibrants (isoprene, $C_5H_8$; α-pinene, $C_{10}H_{16}$; limonene, $C_{10}H_{16}$); panel b) sample humidity dependency of the calibrants**

**4.2.1 Propene-assisted ammonia ion generation**

Ammonium chemical ionization has been successfully used to ionize a wide range of oxygenated VOCs and is particularly suited for the detection of oxygenated species (Zaytsev et al, 2018). Ammonium primary ions were generated by flowing UHP nitrogen over a liquid reservoir filled with a 0.1 vol% ammonium hydroxide solution. Propene has an ionization potential of 9.73 eV and a proton affinity of 752 kJ/mol (NIST Chemistry Webbook, 2021). Ammonia has an ionization energy of 10.1 eV, lower than the

higher-energetic photons emitted by the lamp. Therefore, a mixture of water vapor and ammonia in nitrogen can generate $NH_4^+ \bullet H_2O$ reagent ions (Zaytsev et al, 2019). However, by increasing the volume mixing ratio of ammonia to a level where total primary ion currents are sufficiently high (approx. $> 10^7$ cps), the formation of the less ideal reagent ion $NH_4^+ \bullet NH_3$ becomes dominant. Since the PA of propene is lower than that of ammonia (proton affinity: 854 kJ/mol), proton transfer reactions can form ammonium reagent ions:


$$C_3H_6^+ + NH_3 \rightarrow NH_4^+ + C_3H_5,$$ (10)

Note that H₃O⁺ ions will not form, since the proton affinity of water (691 kJ/mol, NIST Chemistry Webbook, 2021) is lower than the one of propene and the ratio of ammonia to water vapor will determine the ratio of $NH_4^+ \bullet H_2O$ to $NH_4^+ \bullet NH_3$. Table 3 shows the individual primary ion intensities with and without the addition of propene as a dopant; For the same ammonia-water-nitrogen mixture, an overall 17-fold increase in primary ion intensities is observed. The reagent ion cluster distribution is controlled by

ramping the drift voltage as shown in Figure 7. Being able to precisely control the ion energetics and subsequently the reagent ion cluster distribution is a key advantage of a drift tube compared to an IMR without ion guiding electronics.

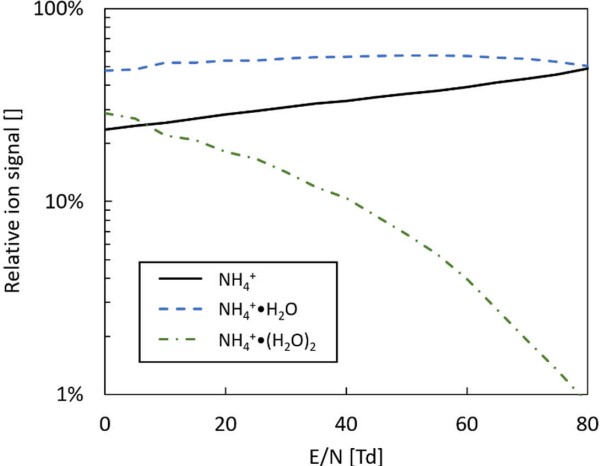

**Figure 7: Reagent in cluster distribution in ammonia mode as a**
**function of reduced electric field (E/N). Drift tube pressure: 2 mbar;**
**drift tube temperature: 60 °C.**

**Table 3: Primary ion intensities in ammonia-mode comparing using a pure**
**ammonium hydroxide solution to a propene-doped solution**

| | Ion intensities [10⁶ Hz] | | |
|---|---|---|---|
| Ion formula | No added propene | 1 sccm added propene | Enhancement factor |
| NH₄⁺ | 0.14 | 5.61 | 40 x |
| NH₄⁺•H₂O | 0.78 | 8.75 | 11 x |
| NH₄⁺•NH₃ | 0.38 | 8.04 | 21 x |
| Total | 1.30 | 22.40 | 17 x |


## 5 Summary

We coupled a custom-designed VUV ion source to a commercially available Vocus Proton Transfer Reaction Mass Spectrometer. This ion source is designed to be compatible with the standard hollow-cathode glow discharge source included with the commercially available Vocus instrument. This allows for operation in both positive and negative ion VUV mode as well as easy

conversion back to the original $H_3O^+$ ion source. In this work, we demonstrate the flexibility of this VUV ion source through the generation of iodide ions, benzene cations, and ammonium cations. The reagent ion cluster distribution can be optimized for any given reagent ion species by adjusting the voltage across the Vocus drift tube. This capability provides improved control over target analytes compared to CIMS instruments with a standard IMR that does not support applying an electric field to the IMR.

The lower drift tube pressure (9 mbar) allows the instrument to operate over a wide range of ambient pressures (50 to 1000 mbar, ground level to an altitude of approx. 20 km). These lower pressures reduce secondary ion chemistry present at elevated ozone levels, an effect that complicates quantification in standard iodide CIMS instrumentation. Additionally, the low operating pressures of the Vocus drift tube enhance the benefits of water addition by removing the sensitivity dependence on ambient humidity. Despite a significant reduction in operating pressure, we present sensitivities using iodide reagent ions that are comparable to other iodide

CIMS instruments using both VUV ion sources as well as standard radioactive ion sources. An additional advantage of this ion source design, particularly for field deployment, is the 10 - 100-fold reduction in consumable gases required for iodide ion generation, using only 10-20 sccm flow of 1000 ppm $CH_3I$ in nitrogen.

Coupling this reduced pressure, bipolar VUV ion source to a Vocus instrument is particularly well suited for conducting high

altitude observations in the UT/LS and addresses the measurement challenges associated with low ambient pressures and a complex sample matrix. While designed specifically for the Vocus instrument, this ion source design is readily adaptable to most CIMS instruments. This work shows that low pressure operation of iodide ion CIMS using a VUV ion source provides significant advantages to more traditional modes of operation, without sacrificing the sensitivity and flexibility of this technique.

**Author contributions:**

AN, AR and PV initiated the project. MB, AN, AR and PV designed the experiment, MB and GN carried them out. MB and GN prepared the manuscript with contributions from all co-authors.

**Acknowledgements:**

M. Breitenlechner and G. Novak acknowledge support from an Earth's Radiation Budget grant, NOAA CPO Climate & CI #03-01-07-001. This work was supported in part by the NOAA Cooperative Agreement with CIRES, NA17OAR4320101.

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
