# Peer review of "A Versatile Vacuum Ultraviolet Ion Source for Reduced Pressure Bipolar Chemical Ionization Mass Spectrometry"

_Atmospheric Measurement Techniques, 2021_

## Referee Comment (RC1)

The paper is well written and covers a topic of great interest for updating the ion sources on CIMS. With the inclusion of the brief tests of the cations the paper is even more pertinent. Recommend publishing with the following issues to be addressed.

Line 29: isobars should be isotopes

Line 36: Unclear sentence 'Relative to nitrate...' Nitrate CIMS cannot measure amines and highly oxygenated organics

Line 88: units of pressure in formula is Pa while the rest of the paper is mbar. Should change to mbar

Line 178 & 229: Is 18sccm of water vapor really added or 18sccm of saturated N2 added?

Line 315: Depending on the C3H6 to NH3 concentration if both are added to the tube illuminated by the VUV lamp, the reaction scheme could be:

1) $C_3H_6^+ + C_3H_6 >> C_3H_5 + C_3H_7^+$
2) $C_3H_7^+ + NH_3 >> C_3H_6 + NH_4^+$

As the PA of C3H5 (736) is less than that of C3H6 (751).

**Scientific significance:**
The paper is if great significance now with the continued development of a replacement for the radioactive ion sources used previously. Not only are they safer, but even generate higher sensitivities.

**Scientific quality:**
Experiments in the paper are well done with a very good description of the instrument used.

**Presentation quality:**
Well done and no recommended changes.

---

## Author Comment (AC1)

**Response to reviewer #1:**

We thank the reviewer for the review and suggestions. The reviewers' comments are written in blue and our point-by-point responses are in green.

The paper is well written and covers a topic of great interest for updating the ion sources on CIMS. With the inclusion of the brief tests of the cations the paper is even more pertinent. Recommend publishing with the following issues to be addressed.

Line 29: isobars should be isotopes

> Our high-resolution time of flight mass spectrometer can resolve both isotopes and isobars, and the isobars (molecules with the same nominal mass, but with different formulae) cannot be resolved with lower resolution instruments. We prefer to use isobars to accurately represent the capability of this instrument.

Line 36: Unclear sentence 'Relative to nitrate...' Nitrate CIMS cannot measure amines and highly oxygenated organics

> Nitrate was indeed used to detect highly oxygenated organics and amines, specifically dimethylamine, as shown in the paper we cited in the original text: Simon et al., 2016.

Line 88: units of pressure in formula is Pa while the rest of the paper is mbar. Should change to mbar

> Thank you for pointing this out, and we have changed the units to [mbar]

Line 178 & 229: Is 18sccm of water vapor really added or 18sccm of saturated N2 added?

> There is no N2 added through the water reservoir, and 18 sccm is entirely water vapor. The low operating pressure allows the addition of pure water vapor from the headspace of a water reservoir at room temperature (water vapor pressure at 22 degC is approx 26 mbar). We use the same delivery system as standard $H_3O^+$ instruments with a flow controlled by a low delta P MFC that comes with the Vocus PTR-TOF.

> Line 139ff now reads: "A separate inlet port for humidification supplies pure water vapor from the headspace of a liquid water reservoir to the sample gas, typically between 10 and 20 sccm, using the low-$\Delta p$ mass flow controller (BRONKHORST HIGH-TECH B.V., the Netherlands) originally shipped with the Vocus PTR-TOF."

> Line 178 now reads: "...the drift tube pressure was 9 mbar, with 18 sccm of pure water vapor added." (Added the word pure);

> Line 229 now reads: "...while 18 sccm of pure water vapor was directly added to the drift tube."

Line 315: Depending on the C3H6 to NH3 concentration if both are added to the tube illuminated by the VUV lamp, the reaction scheme could be:

1) C3H6+ + C3H6 >> C3H5 + C3H7+

2) C3H7+ + NH3 >> C3H6 + NH4+

As the PA of C3H5 (736) is less than that of C3H6 (751).

That is very true. We updated the mechanism. Thank you!

Scientific significance: The paper is if great significance now with the continued development of a replacement for the radioactive ion sources used previously. Not only are they safer, but even generate higher sensitivities.

Scientific quality: Experiments in the paper are well done with a very good description of the instrument used.

Presentation quality: Well done and no recommended changes.

---

## Author Comment (AC2)

**Response to reviewer #2:**

We thank the reviewer for the review and suggestions. The reviewers' comments are written in blue and our point-by-point responses are in green.

Using a VUV source to produce ions at low pressure for chemical ionization atmospheric analysis is a significant innovation that will prove useful to the field measurement community. This manuscript detailing these methods is well-written and organized. I recommend it be published after the authors address the following minor items. Broadly speaking, the manuscript should include additional detail on the techniques presented.

Throughout the manuscript: counts per second in a TOF are useless numbers without providing an extraction frequency. CPS are useful to contrast with quadrupole instruments and other analyzers, but the extraction frequency at which they are acquired in a TOF must be stated clearly in the text AND in major figures such as Table 2. If the cps/ppt values have been scaled to a shorter extraction frequency, this especially needs to be clarified.

> We agree that an extraction frequency should be stated to clarify the meaning of ion count rates. We acquired all data presented with an extraction frequency of 20 kHz, resulting in a mass range of 0 to 780 Th (HTOF with 3kV drift voltage).

> We added a footnote on page at (line 221): "All spectra presented in this work were obtained at an extraction frequency of 20 kHz, resulting in a mass range of 0 to approx. 780 Th."

Table 2: A 10 s limit of detection is not particularly useful for an aircraft instrument. What is the LOD at 1 s? This should be listed in the table, as well.

> Thank you for the comment. For our intended purposes, measurement in the stratosphere, we felt that 10s LODs were appropriate as we do not anticipate structure on the fine scale one expects from sources in the troposphere. However, we do agree that it is of more use to a broader audience to convey 1s detection limits. We have changed table 2 to convey 1s LoDs.

L178: The Vocus PTR source typically operates at a range of 1 to 5 mbar. What modifications were necessary to reach an operating pressure of 9 mbar? Did the authors replace the capillary or skimmers? These details should be added to the manuscript.

> Yes, we changed the skimmer pinhole; original ID was 2.0 mm (measured); we replaced it with an ID of 1.0 mm. We thank the reviewer for identifying the missing information as it is indeed crucial to increasing the operating pressure. The following sentence was added at the end of section 2.3: "A smaller skimmer pinhole (original ID: 2.0 mm) with an ID of 1.0 mm was installed for operation at 9 mbar."

Discrepancy between Figure 2 calculations and Table 1 reported values: It's impossible for the reader not to compare the numbers in Table 1 and Figure 2. I understand that instrumental factors will account for the different I/IH2O- ratio, but I think that the authors should make the minimal effort to explain this by stating the operational parameters for the BSQ amplitude and frequency, and the magnification interface (if used), and add a transmission curve. This is something that they've likely done already or will need to do for a field deployment. Then the numbers from Table 1 should be added to Figure 2b as a scalar for comparison.

In the original manuscript, the following paragraph was intended to explain the discrepancy (line 179ff):

Note that observed cluster distributions are expected to differ from the distributions in the IMR because the mass spectrum is affected by voltages in the ion transfer region between the drift tube and the mass analyser. These voltages were optimized for maximum ion intensities of product ions (which have stronger binding energies than the $I^- \bullet H_2O$ cluster) rather than being optimized for minimum collision activation of ions and avoiding de-clustering in this region.

Yes, it is also influenced by BSQ, specifically. We added this information:

Amplitude: 350V, 1.31 MHz

Because we rely on direct calibration of the CIMS instrument with standards it is not necessary to determine the transmission curve of the instrument, and as such we do not have that information. Our consideration of the mass dependent transmission of the instrument is limited to reducing the total ion current by filtering out 'low' masses such as the primary ions I- (m/z 127) and IH2O- (m/z 145), while maintaining signals at higher masses that include the analytes of interest such as IHCOOH- (m/z 173). This is done to increase the life of the detector as well as the preamp which can be, and has been, damaged by the high-count rates achievable using this method.

The reduction in quantitative humidity dependence is impressive, but it's not immediately clear to me why Br2 sensitivity was shown to go up with increasing RH while in this work it weakly goes down. Could the authors comment on why this might be the case?

Thank you. Unfortunately, we have no mechanistic explanation for this phenomenon at this time. We have considered changes in ion chemistry, sample flows, and surface effects that may depend on RH, but we aren't yet able to establish which of these (if any) explain the small changes in sensitivity at high RH.

L305: It is unclear to me the details of the propene dopant addition. How was the propene added? How much was added? This is a methods paper, so these details are critical and must be included

The reviewer is correct in pointing out this information is missing and should have been included. We added approx. 1 sccm of 99.8% purity propene. This info was added in the text (line 314)

L315 For clarity, there should be an additional equation before Eq. 10 showing the first step in the reaction which ionizes propane

We appreciate the comment and have added line 316, eq 10